# Exploration of Mental Readiness for Enhancing Dentistry in an Inter-Professional Climate

**DOI:** 10.3390/ijerph18137038

**Published:** 2021-07-01

**Authors:** Judy M. McDonald, Corrado Paganelli

**Affiliations:** 1McLaughlin Centre for Population Health Risk Assessment, Faculty of Medicine, University of Ottawa, 216-600 Peter Morand Crescent, Ottawa, ON K1G 5Z3, Canada; 2Dental School, University of Brescia, 25123 Brescia, Italy; corrado.paganelli@unibs.it

**Keywords:** dentistry, competency, mental readiness, preparedness, performance excellence, training, evaluation

## Abstract

Competencies required for dentistry go far beyond the academic or scientific spheres. They incorporate important mental readiness concepts at its core with an appropriate balance of operational readiness (i.e., technical, physical, mental readiness). The aim of this exploratory study was to investigate the importance of mental readiness for optimal performance in the daily challenges faced by dentists using an Operational Readiness Framework. One-on-one interviews were conducted with a select group of seasoned dentists to determine their mental readiness before, during and after successfully performing in challenging situations. Quantitative and qualitative analyses of mental readiness were applied. Study findings were compared with a Wheel of Excellence based on results from other high-performance domains such as surgery, policing, social services and Olympic athletics. The analysis revealed that specific mental practices are required to achieve peak performance, and the balance between physical, technical and mental readiness underpins these dentists’ competency. Common elements of success were found—commitment, confidence, visualization, mental preparation, focus, distraction control, and evaluation and coping. This exploration confirmed many similarities in mental readiness practices engaged across high-risk professions. Universities, clinics and hospitals are looking for innovative ways to build teamwork and capacity through inter-professional collaboration. Results from these case studies warrant further investigation and may be significant enough to stimulate innovative curriculum design. Based on these preliminary dentistry findings, three training/evaluation tools from other professions in population health were adapted to demonstrate future application.

## 1. Introduction

### 1.1. Performance Competencies

Core competencies enable a practitioner to adapt to new situations and perform intricate skills. Dental competencies are focused on knowledge, professional abilities and behaviours. Traditionally, dental education is taught in set increments to produce a dentist with prescribed knowledge [1]. The addition of problem-based learning (PBL) into the dental curriculum enhanced students’ critical thinking, teamwork and general professional competencies. Performance results of Harvard School of Dental Medicine (HSDM) graduates during their postdoctoral training were similar to non-HSDM graduates for traditional residency program competencies. However, the PBL training appeared to provide HSDM graduates with enhanced abilities in independent learning, communication and cooperation skills [2].

In reviewing the Association of Canadian Faculties of Dentistry Competency List, most dental graduates (70%) felt well prepared in most (69%) “bread and butter” aspects of dental practice. They felt less well-prepared in areas such as financial and personnel management, performance of soft-tissue biopsies, management of chronic orofacial pain [3] and certain specialty disciplines [4]. Cultural competency has also been identified as lacking in the dental education community [5,6].

By asking students to rank the overall importance of skills, a dental program could use the information to improve the quality of its offering the students’ preparedness and confidence in basic clinical skills and interpersonal communication. Ultimately, understanding significant core skills was found to be an indicator of better performance [7]. While some core competencies may require extended teaching time, assessment drives the energy allocated by students [7], and therefore important skills must be reflected in the final evaluation.

Research for some time has suggested that the dental profession is over-trained and over-produced for what dentists do and under-trained for what they should be doing, thus advocating for change [8]. Studies of populations having little or no access to dental care show that, despite often poor oral hygiene, most people keep most of their teeth for most of their lives [9]. Marked progress in oral health is largely due to improvements in living conditions and personal hygiene, reduction in smoking and widespread use of fluoride toothpastes, rather than due to the clinical interventions of dentists and insurance-supported industries [10,11]. It is a reminder that current practices are not always right.

It is equally important to acknowledge the La Cascada Declaration [12] generated from a think tank of dental elders who took a similar anti-industrial stance. Although this Declaration may not be fully accepted, it does urge dentistry to be in the forefront for good health and well-being, and become a specialty of medicine, just as ENT (ear, nose and throat) or dermatology. As such, it suggests that oral healthcare be redefined with: oral-health physicians, primary healthcare dental workers and public health dentists. 

### 1.2. Shared Inter-Professional Practices

The World Health Organization (WHO) defines inter-professional education (IPE) as occasions “when two or more professions learn about, from and with each other to enable effective collaboration and improve health outcomes.” IPE and effective collaborative practices are essential for the everyday dental business to be part of the national and global political dialogue. IPE encourages dentistry to develop and implement competencies that support collaborative, high-quality healthcare for patients, families and communities [13].

WHO has highlighted successful inter-professional collaborations at hospitals in Lausanne (Switzerland) and Wales (United Kingdom). At the University Hospital of Lausanne an oral-health team, comprised of dentists, dental hygienists, assistants and technicians, worked with physicians to improve patient health [13] (p. 35). Similarly, the Welsh Government launched a “Fundamentals of Care” policy which led to inter-professional connections between physicians, nurses, pharmacists, dieticians, and speech and language therapists [13] (p. 38).

Overall, health professions (dentistry, dental hygiene, medicine and nursing) have embraced competency-based thinking which has advanced the literature and changed their respective curricula. Four different examination methods to evaluate students and residents are being used across these disciplines—qualitative, quantitative, practical and self-evaluation [5,14]. Evaluation instruments are complex and must integrate concepts, practices and clinical performance. Barriers to developing new instruments include: consensus on core knowledge, misinformation and stereotypical behaviour [5,6]. Given the overlap between professions, all can benefit by sharing inter-professional research and practices [15,16]. Recent trials in the USA and the UK are underway to explore how different professional health profiles can be combined.

### 1.3. Mental Readiness for Dentistry

Nursing competencies were once criticized for being based on conceptual analysis or practitioners’ direct reports. Competencies derived from these methods were said to be either subject to bias or were unable to identify the essential elements for effective care [17]. In 2000, nursing competencies were investigated through realistic, behavioural indicators during successful and unsuccessful events [17]. Ten important competencies were identified in successful events or competent performances—interpersonal understanding, commitment, information gathering, thoroughness, persuasiveness, compassion, comforting, critical thinking, self-control and responsiveness. Conversely, during unsuccessful events or incompetent performances, a lack of self-control and thoroughness were reported most often [17].

Despite the growth in competency research in basic clinical procedures, behaviour (or mental readiness) in dentistry is comparatively under-researched. Dentistry competencies are evolving in areas such as: interpersonal understanding, self-control, thoroughness and cultural aptitude. Dentistry demands high-level training, evolving products, dealing with patient stress and succeeding in business. It is anticipated that skills that help a dentist to be present, focused and manage distractions would be essential for any dentist.

Peak performance training in other fields has been used extensively in preparing top performers to consistently be at their best, optimize conditions and achieve excellence. An “Operational Readiness Framework”, developed by researchers who worked with Olympic athletes, has proven its transferability across very different high-performance settings with equal success: e.g., global health, high-altitude guiding, surgery and policing [18,19,20,21]. Here, individuals perform and make decisions which directly impact lives (and deaths). These studies have identified the critically important direct link between readiness (technical, physical and mental) and performance excellence. The commonality among them is the mental preparedness needed in meeting occupational standards of excellence. Furthermore, mental readiness skills have been found to be a major contributor to peak performance.

### 1.4. Aim

This exploratory study investigated the importance of mental readiness and related success elements for performance excellence in dentistry using the “Operational Readiness Framework,” specifically by:(a)defining challenging situations in dentistry;(b)isolating and evaluating the importance of physical ability, technical knowledge and mental skills in dentistry; and(c)determining and articulating if mental readiness skills are used in daily practices among seasoned dentists.

With this established framework in place, the final objective was to:(d)identify the practical implications (“the fit”) and confirm the capacity for a more comprehensive study of operational readiness with “exceptional” dentists.

## 2. Methods

“Next Generation Risk Science” research highlights the importance of assessing risks and challenging situations using innovative methodologies and addressing them using multiple interventions [22]. The Operational Readiness Framework (Figure 1) was adapted to examine how seasoned dentists perform their best in challenging situations. Every profession has detailed specificity regarding when readiness is imperative and how best to prepare. This framework has been developed to evaluate and create strategies for improving operational readiness in high-risk professions. Individual in-depth interviews were conducted using this empirical approach with experienced dentists. They articulated: challenging situations; readiness (physical, technical and mental); and personal mental-preparedness techniques.

The following is a report that explores if the Operational Readiness Framework can be applied to dentistry. This methodology assesses qualitative and quantitative data in practical ways. Candid quotes were retained to illustrate different applications and to encourage future critical thinking, self-reflection and self-assessment.

Calculation of the relative importance of physical, technical and mental readiness permits job-specific weighting for training, evaluation and curriculum design.

Participants: The interview sample was limited to in-depth interviews with four currently active dentists (three males and one female) derived through referral sampling in the Ottawa, Canada area. All practised general dentistry and one also had dental surgery expertise. Their experience as dentists ranged from 10 to 44 years with a mean of 21 years. Internationally, they had practised across three continents—North America, Europe and Asia. All had dentistry experience in large urban cities and two had practised in rural areas. All had worked with a variety of clients (children, adults, seniors) and one had worked on people experiencing homelessness. Various types of facilities were represented from free government clinics, to hospitals to private clinics. Two were clinic owners.

Instruments: Data were collected using: an introductory Powerpoint presentation entitled *Operational Readiness in High-Risk Occupations*, an *Operational Readiness Interview Guide* and tape-recording and/or notetaking to capture qualitative and quantitative data. As part of university study practice, this initial exploratory phase was constructed in an ethical manner in accordance with strict standards. An information letter was provided to each participant, consent forms were signed and collected, and participants were free to not answer any questions found to be uncomfortable.

Data Collection: This investigation was a qualitative and quantitative collection of data through in-depth interviews from 90 min to three hours in length with seasoned dentists. Detailed interviews afforded the advantage of comprehensive viewpoints and unexpected discoveries to be incorporated into the results. This exploratory study design and procedures were constructed in accordance with the standards ethically approved for previous research involving the same author that examined “mental readiness” in various domains such as surgery, police and Olympic athletes [20,21,23]. The interview related specifically to concentration, mental rehearsal, crisis management and a development of specialized thinking patterns. Based on early consultation with a retired dentist, some changes were made to the questions, although the body of the instrument remained the same. As examples: “dentistry” simply replaced policing/surgery/athletics for open-ended questions; and “dental association/regulatory body” was used to refer to “the governing body.” The interviews were one-on-one, face-to-face, one-time only, and arranged at the dentist’s convenience in a meeting location of the dentist’s choosing. Each interview combined tape-recording and note-taking, with the permission of the dentist. Following completion of the interviews, typed-verbatim interview transcripts were prepared. All dentists received a thank you note for their participation, and a copy of their interview transcript so they could approve its authenticity.

Data analysis: The approved transcripts were prepared for descriptive analysis, as in the procedures used by Orlick and Partington [23]. Participants’ demographic data and interview responses were summarized through various statistical analyses such as qualitative measures of overall operational readiness, and quantitative measures of consistency and averages. Descriptions of each success element along with representative quotes were then given to an independent reviewer. All the data were again reviewed to determine whether or not each success element was evident for each subject. There was full agreement among the reviewers with respect to whether or not a particular dentist’s response warranted assignment to major success elements.

### Controlling Investigator and Information Bias


All questions were treated in a standard way and in a particular order. Given the retrospective nature of the questions, the timeframe for subjects’ recall was kept to a minimum (within the last few months). Questions divided events into segments so details could be easily recalled.The investigator was experienced in answering queries and rephrasing questions when necessary to ensure participants understood before responding. Detailed interviewing was considered superior to a survey for collecting comprehensive information in a new area. General dentistry was captured through representation of both genders, a range of client- and facilities-types, and broad international experiences.Efforts were made to encourage honest, candid responses. The fact that the investigator was not a dentist, created an obvious non-judgmental atmosphere for participants to speak freely. All participants were given time to review a manuscript of their responses to ensure that it authentically captured the experience related to each question.Thematic analysis was performed on the data by an independent reviewer. A cross-checking procedure was used to independently assess the data with methodological rigour.Concurrent with previous studies [20,21,23], it was concluded that manipulating the responses of high-level achievers would be extremely difficult, since these performers tend to be self-directed and act according to their own principles. They are also very keen to articulate and to pass along relevant, realistic experiences to others in the field.


## 3. Results and Discussion

The Operational Readiness Framework was used to: detail the specific challenging operational situations; determine the importance of readiness for performance excellence; and assess if the mental readiness success elements from the “Wheel of Excellence” [24] exist among all four experienced dentists.

### 3.1. Challenging Operational Situations in Dentistry

Using categories from a generic template [21], daily challenges for these seasoned dentists were identified. An extensive list within five categories of challenging situations was compiled (see Appendix A that included: particular patient behaviour (from distracting behaviours to medical complications); degree of difficulty/risk (medically and legally); degree of complexity (from many simple tasks to relieving pain); teaching/managing responsibilities; and special-relationship pressures (with family members or patients). Below is a sample of challenges within each category (for a detailed list, see Appendix A: Personal Profile of Challenging Situations in Dentistry).
Particular patient behaviour—child crying and kicking (not necessarily in pain) ∙ patient’s unbelievably picky husband ∙ those really afraid of the needle ∙ patients experiencing homelessness (a love-hate relationship) ∙ cancer patient on medication with low white blood cell count.Procedural difficulty/risk—extraction of third molars ∙ bone grafts ∙ implants ∙ torn tongue ∙ being tired at end of the day ∙ lawsuits—the number one fear of dentists ∙ complications ∙ anything where you might permanently lose the tooth ∙ matching the colour and angulations on two front teethDegree of complexity—minor things are not hard but complex things like opening flaps or surgical extraction of teeth ∙ mouth full of blood (is patient in pain?) ∙ sustaining proper sterilization practices ∙ equipping clinic with up-to-date first aid ∙ hazardous waste disposal ∙ lack of pharmacological knowledgeTeaching/managing responsibilities—staff issues (e.g., lateness, illness, computers, staff-client problems) ∙ business part (versus just the technique part) is huge! ∙ asking for money if client is unhappy ∙ dealing with insurance telling you what to do ∙ having balance in my lifeSpecial relationship pressures—relatives or a family member (‘relativeoma’) of the patient or friends (‘privatitis’) ∙ multiple relationships to manage (dentist and staff/client/patient) ∙ the private-space thing for some dentists ∙ you start to hurt the patient from not having the person frozen when doing a procedure

Dentists shared candid information related to each type of challenge, for example:

Insurers, the “right thing”—*You can be in situations where the insurance company is telling you what to do versus doing what you think is right and has to be done. Most of the time, the guy sitting behind the desk approving the claim is not a dentist.*


### 3.2. Readiness for Performance Excellence

In dentistry, as in other high-performance occupations, readiness contributes to success and disappointment both in the results (technical and aesthetical) and in the dentist-patient experience.

The dentists defined a “successful” performance as:No complications at 6-month follow-up;No immediate or 24-h post-pain;Patient was comfortable in the clinic;When everything goes well;When complications were dealt with successfully;A positive final result.

They defined a “disappointing” performance as:A first-time procedural error;Final result achieved without profit;Something out-of-your-control;Something simple was missed like taking an X-ray or a tooth wiggling;A complication in the last half-hour of the day;Failure to educate the patient;A patient’s non-compliance.

Overall readiness is based on the balance of physical ability, technical know-how and mental skills.

Readiness in dentistry was defined as:Physically—healthy and fit; has good ergonomics positioning (no stress-related injuries; e.g., back, neck, shoulders, arms, carpal tunnel); has good vision; manages fatigue; performs necessary hand-skill coordination, tactile sensitivity and manual dexterity; is artistic; carves nicely; uses mirror.Technically—has knowledge and application of dental techniques; knows and follows directives/laws; coordinates patient treatment plan and referrals; has good business practices (policies, billing, receivables); has modern equipment and instruments; effective at verbal and written communication.Mentally—confident; committed and compassionate; able to build relationships; clear-headed; adaptable to change; sets a goal and a plan; has a positive attitude and deals well with unplanned situations; able to concentrate and refocus; evaluates decisions with effective coping skills.

Dentists were asked to grade the relative importance of physical, technical and mental readiness in the overall success of dentistry (where the total was 100%). Mental readiness was said to contribute almost half (49%) compared with technical (28%) and physical (23%) readiness (Table 1). Even with a modest interpretation, mental readiness clearly played a major role for excellence with these dentists.

### 3.3. Physical and Technical Readiness

Dental surgical competence implies knowledge of the biological, physiological and anatomical principles involved to safely complete the procedure and communicate properly with the patient to ensure informed consent for treatment. Competence requires minimal surgical trauma, proper tissue management, correct incisions, elevation and reflection prior to actually performing a surgery. Being a fully functioning dentist embraces a broad range of competencies and best practices [25] (DentEd, 2010) captured under:Prerequisites for competences;Patient Examination Assessment and Diagnosis;Communication and Patient Education;Ethics and Jurisprudence;Treatment;Medical Emergencies;Practice Management.

A dentist deemed to be “clinically competent,” for example, in excising an impacted root tip from the alveolus, implies technical knowledge and physical surgical ability specific to the treatment [25] (DentEd, 2010). However, some traditional treatments are undergoing dramatic changes requiring the integration of “soft skills.” Tele-dentistry is possibly a more sustainable way to perform some procedures and business in the future. For example, in orthodontic treatment by distance, a patient receives a set of plastic devices and changes them every 15 days. The dentist must therefore teach and empower patients to check their own fit instead of having repeated visits to assess.

The interviews extracted many candid, sometimes controversial, opinions that shape physical and technical readiness in dentistry such as:
Ability, injury—*The physical part is your hand skills, tactile sensitivity and manual dexterity combined with your health or sore back or whatever. The big injuries and pain from physical stress to avoid are in the back, neck, shoulders, lower-arms, knees, headaches, carpal tunnel and repetitive stress*.
Artistry—*The dentist has to be artistic in order to contour the tooth. I’ve seen a lot of colleagues who, when they first got into dental school, would have never thought that. Some people just don’t have the ability to draw nicely or carve nicely. So obviously, the outcome is not as good as they wished. However, there is no way around it*.
How much?—*The business aspect of dentistry was not given enough in dental school. You cannot invest too much or you’ll have to charge too much. Not all patients can afford it and you might lose business. The business part (versus just technical) is HUGE! You have to win your customers not only by your profession—your excellence in dentistry—but also by having a good one-on-one relationship with them. You have to be positive and you have to be convincing*.

In high-risk, high-performance professions, physical and technical readiness practices are typically well defined, measured and evaluated. In contrast, mental readiness practices may exist but require further isolation, definition and emphasis. For this reason, mental readiness was well defined using an empirical framework (i.e., Operational Readiness Framework, McDonald, 2006) [21].

### 3.4. Mental Readiness Success Elements

Each dentist was asked about their mental readiness before, during and after successful and disappointing performances. The seven success elements on the “Wheel of Excellence” [24] (Orlick, 2003) were used as the base—commitment, confidence, positive imagery, mental preparation, focus, distraction control, and evaluation and coping. All seven were identified as important daily dental practices for performance excellence. Each of these mental success elements are detailed below.

#### 3.4.1. Commitment

Reminders of one’s commitment is especially important to sustain office wellness, motivation and productivity. Commitment is judged by the quality and quantity of work performed. Dentists are routinely pressured to: meet fiscal goals; answer insurance inquiries; keep pace with the patient flow; determine if there is enough time for an emergency procedure; respond when the hygienist asks to check the “unusual;” resolve the ultrasonic equipment that stopped working; give some time to a dental sales rep; and then return home to a quality life with family and friends. For these seasoned dentists, the sources for their continued drive and dedication were simple, pragmatic and described as:Having compassion to reduce pain;setting high standards (e.g., prepared to include “green” and cultural accountability)feeling overall enjoyment;persisting through complexity;feeling responsible for your patients;remaining sincere to avoid complaints and lawsuits;creating a strong work–life balance.

Integrity, ethics and responsibility to patients can be impeded. Externally, for example, COVID-19 imposed modifications to meet the demands during a pandemic [26,27] (Quinn et al., 2020; Chang et al., 2021). Internally, for example, a dentist must decide on the number of billable hours such as “at what age do you restore everything or at what stage do you change a plan in-progress and do an implant?”
Integrity redefined—*Dentistry is more fragile now. More dentists choose group practices over sole ownership for greater protection to cope with customers. Integrity needs to include cultural preparedness, environmental accountability, patient comfort and ethics around payment requests. Being too conservative for dentist safety (or defensive medicine) is a bad practice. A dentist must preserve all aspects of what can be done to help the patient*.
Daily balance—*It is an ongoing challenge to keep your life in balance because unless YOU are right [laugh], nothin’ else is right. I work on a work-play-love-worship model. You need a balance of these things. You can’t work 24 h a day… I try on a daily basis to dedicate myself to each section… If you are aware of that from the start then you’re not likely to get into too much in one area. If you do, at least have the realization why you’re out of balance, and what needs to be done to rectify it. That is HUGE*.

High standards involve vertical and horizontal training. Specialization activities train for a single, very specific purpose (narrow, vertical training). Alternatively, when multiple healthcare streams need the same broader, horizontal training, it would be more cost-effective to have them all take the same training instead of offering it at each profession’s school. For example, European dental schools see this application for complex cases and general-purpose activities such as “green accountability” [28] (Duane et al., 2019).

#### 3.4.2. Confidence

Dentists expressed that they derive confidence from multiple sources such as patients, colleagues, continuous training and personal initiatives. Confidence under stress is a practised skill. In an *“I don’t know how”* moment, albeit a procedure or a possible complication, a dentist can demonstrate a positive mental attitude and confidence by opting to refer. Practices reported by dentists as helping them acquire and maintain confidence were to:Acquire self-confidence through practice;Assess and accept patients who are like-minded;Know your patient;Promote team pride;Evoke the Peter Principle and refer to specialists;Join study-group clubs;Seek mentor support;Pursue continuous education (near and abroad).

In dentistry, confidence is important not only for the dentist but for the staff and the patient.
Three-way pride—*‘If you can’t brag on ‘em, fire ‘em!’ It’s a two-way street. If your staff can’t brag about you, they should leave. And the same with a patient: If a patient can’t brag about you as their dentist, then [spit sound]… they should leave too*.

If future travel is reduced due to pandemics and new border laws, developing through international in-person meetings might be lost. Greater consideration for local support, on-line networking and e-learning may be necessary as a way of gaining confidence and advancing in the field.

#### 3.4.3. Positive Imagery

Imagery or visualization is used before, during and after a critical performance to rehearse, recall and create desired outcomes. It allowed these dentists to mentally see, feel and control anticipated results. The specific applications reported for dentistry were to:Visualize the final result;Draw pictures of the tooth;Rehearse using visual aids;Prepare to treat with an optimistic mental attitude;Develop a positive patient-relationship.

One dentist related it to sport analogies:

Visualize first—*You always try to see the final result in your head. It’s like a golf shot. You want to visualize where the shot’s going to ideally end up, and then you execute it… I’m doing it as I’m talking to them, if I can, because so many things come in to play*. 

#### 3.4.4. Mental Preparation

Dentists seem to systematically develop and refine their habits/practices such as rituals, pre-planning and positive thoughts to enhance their existing abilities going into a challenging situation. These preparations can be days before, the night before, an hour before and/or minutes before. Pre-event practices reported were to:Pre-plan the logistics;Allow ample time for the appointment;Anticipate complications and options;Have daily preparation rituals;Put pictures up to explain the plan or present alternatives to waiting patients;Prepare for the unexpected.

Simple individualized preparation strategies work best such as:
Rituals, calm—*I believe that if you are psychologically calm and relaxed, you can perform better… It is whatever keeps your psyche down—like drinking tea, reading sports results, doing mindfulness or yoga, praying… even taking one deep breath*.
Retrench—*Take a deep breath. Start with that. Retrench. Sometimes I’ll leave the room if the patient isn’t in any danger. Go back and retrench. Think about it for a couple of minutes. You’ve got that luxury*.

Anticipating complications and solutions is a well-known practice of top performers. By foreseeing what is going to happen, and improving upon it, a dentist can change future outcomes. Take for example, the exploration by early Canadian and American dental pioneers of methods to address one of the chief obstacles to visiting a dentist—the fear of pain. One of Canada’s early top performing dentists, Dr. Jacob Neelands, is credited with introducing to Canada the use of nitrous oxide gas as an anesthetic for the extraction of teeth. This innovation meant that visiting the dentist could be a relatively pain-free experience, thereby improving the preparation and outcome for both patient and dentist for years to come [29] (Royal College of Dental Surgeons of Ontario, 1993).

#### 3.4.5. Focus

Precision in dentistry demands concentration on the task at hand with a sense of flow while maintaining connection with the patient. This mindset was described as the ability to:Have total concentration;Communicate non-verbally;Control your own emotions by staying calm and relaxed;Have a patient-centred mindset (e.g., sharing the steps aloud);Be nice;Be comfortable feeling the rhythm and flow of the procedure (the economy of movement);Create effortlessness in your practice.
Simply be nice—*Dentists are probably one of the few professionals that people will allow to work in very close proximity for about an hour—really close in their mouth, face with your face, bodies very close to each other. Patients have to be really, really trusting of the dentist. It’s not only doing a good job from a technical aspect, you are also being allowed to be close to them. Patients we see every day are mostly in pain and not in a good mood… You have to be nice*.
Get it right—*It’s not just doing the technical. Getting the right relationship with the patient makes things easier. People don’t like the damn needle. If people like you, you can hurt them a bit. Okay, it hurts but if they don’t like you, it hurts a lot more for them [laugh]*.

Concentrating on the procedure, comforting the patient and establishing a flow are complex and not easy to achieve. If dental students had greater understanding of the need for these practices with more detailed feedback during training, these competencies might greatly improve.

#### 3.4.6. Distraction Control

Expected or unexpected interruptions, negative feedback or bad luck can happen before, during or after dental procedures. Strategies are required to get back on track quickly (often so you are undetected) to avoid impacting performance. Multiple refocusing skills were reported to control these distractions such as to:Use creative personal means to immediately control the excitement;Be honest with the patient;Be willing to apologize;Be willing to be flexible;Redo or correct;Put the distraction on hold;Take charge and push through;Avoid mental fatigue by taking a walk, drinking tea or switching tasks.

Control doubts—*Now we don’t use amalgams anymore because of the mercury. It’s always in the back of your mind—‘half of what I’m doin’ is wrong.’ However, you do what you can. You have to have goals. You have to be honest and sincere. Mistakes happen but you did what you could and it’s out of your control*.

Get back on track—*I am reasonably fit and well rested when I go in there so I don’t feel that physical fatigue. It is mostly mental if I start to lose my edge in being sharp. Dentists have to be able to calm down in a way that nobody else notices like using humour, lowering your voice, speaking very slowly or taking time to choose the right words. Slowing down and talking in a calm way actually calms the patient as well*.

#### 3.4.7. Evaluation and Coping

The dentists interviewed emphasized the need to adapt to rapid changes in technology. They also extracted important lessons after a critical procedure to refine their mental approach. Debriefing went beyond self-assessment to include colleagues, team and patients. Their reported evaluation and coping practices were to:Be reflective;Have a realistic goal;Discuss at study group;Invest in continuing education;Recognize and assess technological changes;Seek patient feedback;Do your best;Adjust after an error;Recognize unexpected setbacks;Practice work–life balance.

Perfection—*You can never do it right. Generally, you’re very self-critical of most things you do. You see what could have been better, even though the patient’s not in any particular danger—little tiny things and sometimes bigger stuff—then maybe you start over because you’re not happy with it*.

Amicable—*Absorb their ignorance. You say, ‘Well, I cannot do this because of this and this. Even though I’m telling you this, you might want to get a second opinion.’ You can throw it back. It’s a good release for you. ‘You know what? If you’re going to challenge my judgment, I’d rather not see you’—but you say it in a really nice way*.

The mental health movement has brought attention to “wellness” for frontline, emergency, first-responders. Dentists must find ways to assess and cope with their extraordinary dilemmas that include long hours, financial risk, critical decision-making and the inherent intimacy with patients.

Painful pressures—*There’s a lot of pressures financially and ethically. It’s extremely stressful after five years—it is difficult to expand, make everyone happy or everything else you don’t think about. Some dentists are uncomfortable with the private-space thing—and that’s not good. It can all lead to a lot of psychological problems (like depression and drugs) in looking for an out because it’s so painful*.

Exercise restraint—*Banks come calling offering the world for some immediate gratification which can be appealing after almost a decade of schooling. Hold off on big purchases until the practice is paid off—most often after 10-years. Joining a group practice or associate can still mean overwhelming debt. Debt affects motivation and satisfaction with a ripple-effect on safe and competent delivery of treatment. It’s a lousy way to start what should be a rewarding career-chapter*.

### 3.5. Comparisons across Professions

The high mental-readiness score in dentistry (49%) supports results found with other high-achievers (Table 2) such as in surgery, health research, social work, policing and elite athletics [18,20,21,23,30] (McDonald et al., 1995; McDonald & Gyorkos, 2016; McDonald & Hale, 2021; McDonald, 2006; Orlick & Partington, 1988, respectively). For this sample of professions, physical readiness was ranked highest for Olympians and lowest for surgeons. Technical readiness was ranked second by all but elite athletes.

#### 3.5.1. Surgeons

The profiles for surgeons and the dentists interviewed for this study were less similar than one might expect. Surgeons showed a wide disparity in physical readiness (10%) compared to a more even distribution in their mental (49%) and technical (41%) readiness. Conversely, in dentistry, the more even distribution was between physical (23%) and technical (28%) readiness. The manual dexterity (such as drilling and filling) in performing physical techniques in dentistry relates to any area of surgery but with more visually detectible results. Technically, the need for both business and medical knowledge differentiates dentists from surgeons. Like surgeons, dentists shared identical high mental-readiness scores (49% and 49%, respectively, see Table 2). Dentistry and surgery may align well related to their practice of confidence, visualization and patient preparations [20] (McDonald et al., 1995).

Building confidence—*Surgeons have great belief in the significance of their interventions. The most successful moments in my life are those when I removed a tumor. That experience over the years has led to building-up my confidence*.—Oncology surgeon

Team visualization—*I meet with the surgery team and together we visualize the operation scenarios and discuss the details. This is the most important preparatory step and it really does not matter who eventually performs the surgery*.—Cardiothoracic surgeon

Patient preparations—*Beforehand, you communicate with your patient so they understand what you are going to do and what complications could happen. You see the patient has confidence in you*.—Cardiac surgeon

However, in lifesaving, emergency surgery, it may not matter if the surgeon knows the unconscious patient or not—the surgeon is trying to keep the person alive. Conversely, the dental patient is generally awake with the dentist working in their “private space.” Creating a positive patient attitude is fundamental to a dentist’s simultaneous readiness. Here, it is about aligning personalities.

#### 3.5.2. Social Workers

Profiles for social workers and these dentists were very similar in all three areas—physical, technical and mental readiness. While dentistry involves medical technologies, the practice is more about being suggestive and encouraging patient empowerment and patient decision-making than about explaining the technology. Dentists also focus on de-escalating patient fears. All these competencies are essential in social services.

## 4. Conclusions


*Significance of mental readiness:* Compared to dentists, top performers in other high-risk professions demonstrate excellent technical and physical skills but where dentists stand out is their proficiency in mental readiness [17,18,19,20,21] (McDonald et al., 2016, 2015, 1995; McDonald, 2006; Zhang et al., 2000). Similarly, these case studies indicated that well-honed mental readiness was a major contributor (49%) to best performances in demanding and routine dental procedures. If confirmed, this would allow insight for training, evaluating and benchmarking excellence specific to dentistry. By extension, improving mental readiness could increase productivity and morale in the delivery of quality patient care.*Job-specific operational readiness* is critically important for preparedness. For these dentists, the weighted importance for readiness was: physical 23%, technical 28% and mental 49%. Trainees need to comprehend and progress through these readiness competencies to be fully functioning dentists. Training allocations in dentistry have not changed in 40 years. The curriculum in 1980 was the same as today—10% surgery, 10% orthodontics, 10% periodontics, 10% prosthetics and many small topics at 5% each (e.g., paedodontics, oral mucosa, community dentistry and prevention, endodontics, rehabilitation).*Inter-professional preparedness:* Similarities and differences were evident between health professions such as dental, surgical and social services. COVID-19 has shown us that it *is* possible to shape new activities and reset traditional ways in a changing environment. For example, training in the role of gloves to reduce facial self-touching is a shared strategy across health professions. At a recent US-Europe Congress [31] (Roundtable Discussion, 2021), it was identified that virtual mobility now supports presenting complex and rare cases at international, inter-professional Grand Rounds. All parties would need some common knowledge about each other’s profession (i.e., able to properly convey information in medical doctor’s language and vice versa). Everyone would need to know their role and learn how to communicate well together. Additionally, linking electronic medical records to dental records (i.e., oral cavity and dental database) would be useful and symbolic. Similarly, mental readiness practices may be ideal to share across disciplines such as daily rituals, visualization, de-escalation and patient empowerment.Criti*cal mental practices:* Certain mental practices may hold more importance in some dental procedures. For example, projecting confidence and focusing under pressure may apply best in particular surgeries. Successful mental practices can therefore be isolated, emphasized and evaluated to improve specific clinical competencies.*Adapted**training tools:* Three practical tools found to have beneficial effects in social services and policing [30,32] (McDonald & Hale, 2020; Ontario Police College, 2008, respectively), have been tailored and drafted for dentistry. The adaptation was based on the study findings and review by a subject matter expert in dentistry.


### 4.1. Personal Profile of Challenging Situations in Dentistry

*Instructions for use:* Knowing your strengths and limitations can boost self-confidence. A detailed list of daily challenges identified by seasoned dentists has been created and grouped into five categories. An exercise has been developed for trainees to design a personal profile of these challenging situations. It is intended to assist them in: developing new strategies; refining existing skills; realizing when to refer; finding opportunities to mentor others; and discovering possible areas for specialization in dentistry (see Appendix A: Personal Profile of Challenging Situations in Dentistry).

### 4.2. Operational Readiness Performance Indicators for Dentistry

Thirteen (13) performance indicators were specifically designed to approximate the ratio of job-specific competencies for dentistry found in this study. That is, in addition to developing the technical (31%) and physical (23%) readiness skills required for operational readiness, the necessary isolation and emphasis are placed on developing mental readiness skills (46%). It is important to understand that performance indicators and practices do not need to be equally reflected in the number of lectures, proportion of training or all supplemental assessments. However, in the final assessment of a “fully functioning dentist,” trainees must demonstrate comprehension and competency (pass/fail) in all physical, technical and mental performance indicators. While more time will be spent after graduation on honing a specialty, a foundation in mental preparedness will provide the resiliency and sustainability for facing the inherent complexities and difficulties of the job (see Appendix A: Operational Readiness Performance Indicators for Dentistry).

### 4.3. Clinical Training Assessment for Dentistry Trainees

Consistent with the principles of adult education, trainees are expected to take an active role in their training by maintaining a daily log noting particular instances where they have demonstrated proficiency in the required 13 performance indicators. Clinical training can be divided into roughly three equal stages of trainees’ progress, from comprehension to full competency. If an instructor identifies a problem at any stage, it is discussed so the trainee can be given the opportunity to improve their performance. To ensure objectivity, instructors should back up their assessment with examples (or lack of examples) from the trainee’s and their own daily log (see Appendix A: Clinical Training Assessment for Dentistry Trainees).

## 5. Recommendations


*Conduct a comprehensive operational readiness dentistry study:* These results provide impetus to conduct interviews with wider and different environments, ages and range of dental specialties. Building on this research, a comprehensive, operational readiness study with highly respected dentists would distinguish dental specializations (vertical training) and augment inter-professional teamwork (horizontal training).*Update dental curriculum:* A new method is needed to reassess the weight of different aspects to reduce over-crowding of the fundamentals. While the overhaul of dental schools described in La Cascada Declaration [12] (Cohen et al., 2013) may be too extreme, these detailed, case-study results may lead to changes. As preliminary results, they give cause to re-examine the current dental-school curriculum, the number of dentists trained and the inter-professional possibilities linked to population health needs.*Seek inter-professional training opportunities:* “Traditional” approaches are shifting in dentistry as: graduates choose group versus sole practices; preventive health measures are imposed during pandemics; and frequent travel restrictions occur between borders. Life-work balance, dentist-patient relationships, ‘team’ work and international mentoring opportunities may all require new strategies to succeed. Finding common grounds for inter-professional preparedness may be valuable for dentistry.*Enhance recruitment, training and follow-up:* The recruitment and selection process could also be further refined to prefer candidates who demonstrate critical mental skills needed to face and cope with the complexities in dentistry. Mental readiness can be integrated into existing dentistry learning objectives. As students acquire new physical abilities and technical know-how, mental skills can be integrated into the curriculum and final assessment. A five-year follow-up by dental schools may help to remind new graduates of the full complement of skills needed and to support them in their practice.*Test operational tools:* The goal of clinical training is to develop a competent, independently functioning dentist who will provide dental services in a safe, courteous and effective manner. To achieve this goal, a trainee must demonstrate competence in performance indicators. A clinical trial with current dental students would create a preliminary step to assess pre- and post-exposure to the Operational Readiness Framework through the three draft tools. Student and instructor feedback would give further insight into effective applications of mental readiness to dentistry.


## 6. Measures of Potential Impact

We sought reactions to our draft paper from four stakeholders—the dental student, the seasoned dental hygienist, the experienced dentist and the academic leader. The sample of opinions below highlight a need for:adaptable preparedness strategies for the ever-changing, post-pandemic conditionsearly reflection and practice of “survival” skills in mental readinessmore holistic integration of “soft skills” and operational procedures into dental educationearly inter-professional collaboration to advance learning, leadership and public-health involvement.

### Stakeholder Reactions

Undergraduates—*Since COVID, we are requested to be ready in a new way for dental practice. Each of the seven dimensions for mental readiness must now be assessed with a COVID lens. Evaluation and coping are of particular importance in adapting to our new, rapidly evolving rules. Our new dressing sequence has itself become a preparation ritual. Despite these difficulties, maintaining a positive management style for finding solutions remains critical. Commitment now means working with a new satellite team and operators. We need to redefine our focus and distraction control since human touch now has barriers which circumvent what was once possible. Finally, we must find a way to remain confident while adhering to strictly imposed rules*.—Forum with 10 Dental Undergraduates, Europe 2021

Hygienist—*There is a significant investment that goes into preparing for this profession and many who come out sadly find out that they hate it. Given the intimacy, brevity and sheer responsibility (fiscal, employment, medico-legal, technical), if you don’t like it, it’s going to be a big slog. Early reflection on readiness may help post-grads achieve clearer direction, in addition to happiness, satisfaction and pride in their work. Building confidence, rituals and coping strategies are early survival practices much needed for work–life balance in dentistry*.—Seasoned Dental Hygienist, North America, 2021

Seasoned Dentist—*‘Manual dexterity’ is one performance indicator that contributors to the overall physical-technical-mental readiness needed for a fully functioning dentist. Knowing the appropriate ratio and offering autonomous mentoring quotes can help schools recognize and promote mind–body-spirit balance at the undergraduate level. Schools encourage a life outside of dentistry albeit through music, cooking, etc. Future research is needed to reflect operational readiness and balance not only in general dentistry but also the demands in the specialties (orthodontics, oral surgery, periodontics, paedodontics)*.—Experienced Dentist, North America, 2019

Dean—*An administrator’s job is to watch for trends and appropriately adapt programs. This research reinforces the need for: early integrating of inter-professional teams, re-examining ‘integrity’ and emphasizing mental readiness. Dentistry today is less about developing a team but rather working in teams. Treating a cleft palate or sleep apnea involves the evolution and collaboration across professional teams of speech therapists, neurologists and surgeons. Integrating first-year dental students with advanced nursing students would be an evidence-based approach to promote learning, leadership and a higher involvement in public health*.—Dean of Dentistry, Europe, 2019

## 7. Limitations of the Study


The sampling size of seasoned dentists was small and not intended to make inferences to all dentists. Overall transferability of the Operational Readiness Framework for dentistry was considered more important than generalizability. Case studies permit exploration and description to enhance (in this case) the understanding of mental readiness in dentistry, expand the knowledge base and generate hypotheses for future study [33] (Richardson, 1994).Selection and method bias: Dentists’ free time for research is limited and difficult to access. While referral sampling was required to allow for this preliminary study, it made it impossible to determine the sampling error. At this time, an in-depth, face-to-face meeting was the best method available to explore this under-researched area for this medical specialty.


## 8. Strengths of the Study


The qualitative nature of this research elicited candid conversation to collect comprehensive information and unexpected viewpoints in a new subject matter for dentistry.The study design also incorporated measures to quantify the relative importance of physical, technical and mental readiness for a dentist’s overall preparedness.The track-record across other high-risk professions of this empirical approach may offer innovative, evidence-based direction for change in the dentistry curriculum.


## Figures and Tables

**Figure 1 ijerph-18-07038-f001:**
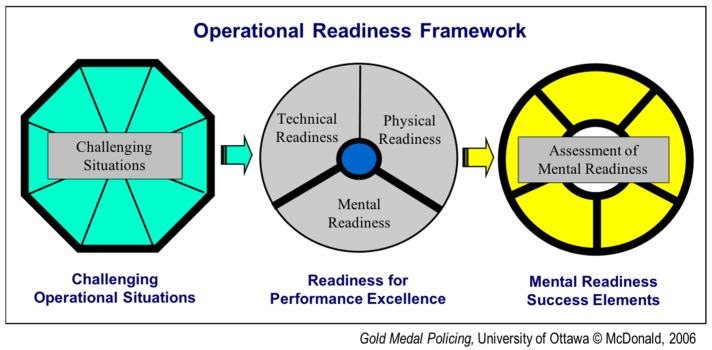
Three stages of the Operational Readiness Framework [21] (p. 2).

**Table 1 ijerph-18-07038-t001:** Importance for Readiness in Dentistry.

Readiness Factors	Mean %	Standard Deviation
Physical Readiness	23	11.39
Technical Readiness	28	13.47
Mental Readiness	49	23.47

Note: On a scale where readiness factors totalled 100%.

**Table 2 ijerph-18-07038-t002:** Interdisciplinary Comparison of the Importance of Readiness for Performance Excellence.

Readiness Factors	Dentists	Surgeons	Health Researchers	Social Worker	Police	Olympians
Physical Readiness	23%	10%	18%	24%	28%	38%
Technical Readiness	28%	41%	42%	29%	32%	20%
Mental Readiness	49%	49%	40%	47%	40%	42%

Note: On a scale where readiness factors total 100%.

## Data Availability

The data are not publicly available due to the privacy and professional specificity of the people who took part in this research.

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
