# Peer review of "Exploration of Mental Readiness for Enhancing Dentistry in an Inter-Professional Climate"

_ijerph, 2021, doi:10.3390/ijerph18137038_

Round 1
Reviewer 1 Report
The assertion in line 59 have to be referenced.
The paper is long. Because it is an issue new to dental education, the descriptions and explanations are long and detailed. It seems to me that to increase readability it should be shortened and somehow simplified, leaving more in depth delving in references.
Author Response
The assertion in line 59 have to be referenced. Completed.
that is "Research suggests that the dental profession is over-trained and over-produced for what they do and under-trained for what they should be doing." I suggest to insert something referring to the need to add new aspects of profession .... Thank you. Reference and qualifying text has been added.
The paper is long. Because it is an issue new to dental education, the descriptions and explanations are long and detailed. It seems to me that to increase readability it should be shortened and somehow simplified, leaving more in depth delving in references. We have simplified the language. If the reader needs more detail in the mental readiness several previous studies (i.e, McDonald, J. & Gyorkos, T. (2016; McDonald et al, 2015; McDonald, 2006; McDonald et al, 1995). Additional criteria requested by Reviewer 2 were added as Appendix IV.
SIMPLE LANGUAGE to increase readability requests (Reviewer 1 & 2) was addressed by haing an independent editor review the manuscrippt. As a result, text was removed; examples were clarified and in one case replaced; and vague terms were explained.

Reviewer 2 Report
Dear authors,
I read your manuscript with great interest.
Using both qualitative and quantitative methods your aim was to investigate and identify elements for performance excellence in dentistry and the importants of mentall readiness in daily practice.
Results of your study suggest that seasoned dentists use the same strategies or "seven success elements" in daily dental practices for performance excellence and value Mental Readiness highly as other High-achievers across professions.
You have acknowledged and commented well on the weaknesses of the study but not highlighted the areas of strenght.
Even if the journal IJERPH does not require a strict format the manuscript must follow and contain sections according to "instructions to authors". This manuscript lacks required sections: Clear Conclusion ("this section is mandatory"), Funding Information, Author Contribution, Conflict of Interest and other Ethical statments. Please consider revising your manuscript according to guidlines.
Specific comments:
Abstract: Line 24-26 the statment "to stimulate innovative curriculum design" is repeated, maybe the sentence can be revised.
Introduction
Last paragraph under Performance competencies, line 67-75. I consider this section more relevant in the discussion part of the study when discussing recommendations and future considerations for innovative curriculum design.
Line 94-96 You state that there are four examination methods to evaluate students, I lack reference(s) to these four methods or the statement.
First paragraph under Mental readiness for dentistry contains only one reference, I wonder if the whole paragraph and statments are based on the same study or others, please clarify.
In the second paragraph under the same section there is the last sentence stating "Having skills that help a dentist to be present, focused and manage distractions are critical, and a goal for any dentist." How do we know that this is true, is there a study done to make this conclusion or statement.
Methods
There is no statment made about the ethical approval of the study.
Line 156 sentence begining "The following is a report ..." I do not understand what do you mean by this statement. Please clarify.
Line 180 , the use of emotianlly charged words "very experienced dentits", recommendation to skip "very" just use seasoned as in the abstract.
Line 187 you state "although some changes were made to the questions to incorporate dentistry vocabulary,..." Please give examples of these changes and make a statement who made these "vocabulary suggestions" do they came from dentists or?
Line 197 statement "... appropriate statistics calculated." Please clarify what you consider appropriate statistics for this study.
Line 208 the word "very" please consider leaving out.
Results and Discussion
First paragraph line 230, please consider replacing "among these experienced dentists" to "among four experienced dentists."
Line 297 First sentence feels missplaced and that it should be moved to Methods.
Line 345-346 Last sentence, statement requires a reference "...using an empirical framework."
Line 350 "the seven success elements" please consider presenting them in parentheses first not just reffere to them as "below".
Page 11, Line 487-491 I have a hard time seeing this quote relevant for this section, maybe more appropriate under "have a realistic goal" or recommendation to skip the first two sentences from the quote and just use quote from "It´s always in the back of yoru mind.."
Suggestion to use quote page 10 line 466-469 as it shows strategies dentist use to calm down and get back on track, avoid metal fatigue.
Page 12 Line 530-535 Quote fits also under Commitment and creating a strong life balance.
Page 12, Table 2. revise collum under Surgeons 10%, 41%, 48% does not ad up to 100%.
Good Luck with the revision!
Author Response
Using both qualitative and quantitative methods your aim was to investigate and identify elements for performance excellence in dentistry and the importants of mentall readiness in daily practice.
Results of your study suggest that seasoned dentists use the same strategies or "seven success elements" in daily dental practices for performance excellence and value Mental Readiness highly as other High-achievers across professions.
You have acknowledged and commented well on the weaknesses of the study but not highlighted the areas of strenght. Thank you. Three areas of strength were added.
Even if the journal IJERPH does not require a strict format the manuscript must follow and contain sections according to "instructions to authors". This manuscript lacks required sections: Clear Conclusion ("this section is mandatory"), Thank you. Recommendations were reformatted into clear Conclusions and Recommendations.
Funding Information, Author Contribution, Conflict of Interest and other Ethical statments. Please consider revising your manuscript according to guidlines. These additional criteria were added as Appendix IV.
Specific comments:
Abstract: Line 24-26 the statment "to stimulate innovative curriculum design" is repeated, maybe the sentence can be revised. Yes, corrected.
Introduction
Last paragraph under Performance competencies, line 67-75 / 75-81. I consider this section more relevant in the discussion part of the study when discussing recommendations and future considerations for innovative curriculum design. Agreed. La Cascada Declaration (Cohen et al, 2013) text has been shortened in the introduction and reinforced in the Recommendations re innovative curriculum design.
Line 94-96 / 100- 102 You state that there are four examination methods to evaluate students, I lack reference(s) to these four methods or the statement. Agreed. References were inserted
First paragraph under Mental readiness for dentistry contains only one reference, I wonder if the whole paragraph and statments are based on the same study or others, please clarify. Yes, Zhang et al reviewed 7 studies and then conducted a new study in 2000. To clarify, references were inserted for both points.
In the second paragraph under the same section there is the last sentence stating "Having skills that help a dentist to be present, focused and manage distractions are critical, and a goal for any dentist." How do we know that this is true, is there a study done to make this conclusion or statement. Agreed. No study has been done thus intended to emphasize the need for this research. Claim has been removed and worded as: “It is anticipated that…”
Methods
There is no statment made about the ethical approval of the study. A brief ethics statement was added to the Data Collection, as well as a detailed “Institutional Review Board Statement” as Appendix IV Both explain the exploratory nature of the work that followed previous studies.
Line 156/195 sentence begining "The following is a report ..." I do not understand what do you mean by this statement. Please clarify. Agreed. Sentence was edited to be clearer. “The following is a report that explores if the Operational Readiness Framework can be applied to dentistry.”
Line 180 / 218 , the use of emotianlly charged words "very experienced dentits", recommendation to skip "very" just use seasoned as in the abstract. Changed as per recommendation.
Line 187 / 226 you state "although some changes were made to the questions to incorporate dentistry vocabulary,..." Please give examples of these changes and make a statement who made these "vocabulary suggestions" do they came from dentists or? Statement added that a retired dentist was consulted with two examples of changes provided.
Line 197 /263-265 statement "... appropriate statistics calculated." Please clarify what you consider appropriate statistics for this study. Details of qualitative and quantitative measures have been added.
Line 208 /275 the word "very" please consider leaving out. Removed.
Results and Discussion
First paragraph line 230 / 297, please consider replacing "among these experienced dentists" to "among four experienced dentists." Replaced with “among all four experienced dentists.”
Line 297/ 379 First sentence feels missplaced and that it should be moved to Methods. Statement was reduced to introduce the question. Quantitative results were described in the methods as per your comment above.
Line 345-346 / 431-432 Last sentence, statement requires a reference "...using an empirical framework." Reference added.
Line 350 / 435 "the seven success elements" please consider presenting them in parentheses first not just reffere to them as "below". The sentence structure was changed to list the seven success elements.
Page 11, Line 487-491 / 615-618 I have a hard time seeing this quote relevant for this section, maybe more appropriate under "have a realistic goal" or recommendation to skip the first two sentences from the quote and just use quote from "It´s always in the back of yoru mind.." Good idea to remove first line. Quote was shortened and re-titled “control doubts” to emphasize controlling a distraction by refocussing on the goal and what is within your control.
Suggestion to use quote page 10 line 466-469 / 619-623 as it shows strategies dentist use to calm down and get back on track, avoid metal fatigue. Agreed, moved and retitled “get back on track.”
Page 12 Line 530-535/ 473-478 Quote fits also under Commitment and creating a strong life balance. Agreed and moved to Commitment.
Page 12, Table 2. revise collum under Surgeons 10%, 41%, 48% does not ad up to 100%. Thank you for catching typo.
SIMPLE LANGUAGE to increase readability requests (Reviewer 1 & 2) was addressed by haing an independent editor review the manuscript. As a result text was removed; examples were clarified and in one case replaced; and vague terms were explained. Adeed research declaration in Appendix IV with author contribution, funding, ethics statements. We have clarified introduction, methods and conclusions.

Reviewer 3 Report
The objective should be revised to the role of gloves in the reduction of facial self-touching to make their conclusion
Author Response
The objective should be revised to the role of gloves in the reduction of facial self-touching to make their conclusion
The suggestion was inserted under inter-professional preparedness shared by surgeons and dentists (Conclusion #3)
Language was simplified for a smoother read to increase readability requests was addressed by haing an independent editor review the manuscript. As a result text was removed; examples were clarified and in one case replaced; and vague terms were explained
